# Mapping Wetland Types in Semiarid Floodplains: A Statistical Learning Approach

**Megan Powell** [1,2] , **Grant Hodgins** [1], **Tim Danaher** [1], **Joanne Ling** [1], **Michael Hughes** [1,3] **and Li Wen** [1,2,*]

1 Water and Wetlands, Science Division, Office of Environment and Heritage, NSW 2000, Australia; megan.powell@environment.nsw.gov.au (M.P.); grant.hodgins@environment.nsw.gov.au (G.H.); tim.danaher@environment.nsw.gov.au (T.D.); joanne.ling@environment.nsw.gov.au (J.L.); michael.hughes@environment.nsw.gov.au (M.H.)
2 Department of Environmental Sciences, Macquarie University, Sydney NSW 2109, Australia
3 School of Earth Atmosphere and Life Sciences, University of Wollongong, North Wollongong NSW 2500, Australia
* Correspondence: li.wen@environment.nsw.gov.au

**Abstract:** Detailed vegetation maps are needed for wetland conservation and restoration as different vegetation communities have distinct water requirements. It is a continuous challenge to map the distribution of different wetland types on a regional scale, and a trade-off between the categorical details and availability of resources to ensure broad applications is often necessary for operational mapping. Here, we evaluated the capacity and performance of statistical learning in discriminating wetland types using Landsat time series and geomorphological variables computed from Light Detection and Ranging (LiDAR) and Shuttle Radar Topography Mission (SRTM) digital elevation model (DEM). Our study showed that there was a discrimination limit of statistical learning in wetland mapping. The approach was clearly inadequate in distinguishing certain wetland types. In semiarid Australia, our results suggested that the appropriate level for floodplain wetland mapping included four classes: tree-dominated woodlands, shrublands, vegetated swamps, and non-flood-dependent terrestrial communities. Our results also demonstrated that the geomorphological metrics significantly improved the accuracy of wetland classification. Furthermore, geomorphological metrics derived from the freely available coarser resolution SRTM DEM were as beneficial for wetland mapping as those extracted from finer scale commercially-based LiDAR DEM. The finding enables the widespread applications of our approach, as both data sources are freely available globally.

**Keywords:** statistical learning; geomorphological metrics; predictive power; floodplain wetland types

## 1. Introduction

Wetlands provide a range of key ecological services including maintaining regional and global biodiversity [1], controlling flooding and diffusive pollution [2], and sequestering carbon [3]. Increasing recognition of the ecological importance of wetlands has resulted in a growing interest in mapping their distribution [4,5], and a range of wetland maps was developed at the global [6,7], the continental and regional [8,9], and the catchment and local scales [10,11]. The methods by which wetlands have been identified or classified have varied, and the results have often been incompatible or inconsistent within and across spatial scales [6,12,13].

It is widely recognized that ecological services are closely associated with wetland types; for example, alluvial floodplain swamps are efficient at nitrogen processing [14], and isolated forest

swamps have the capacity for carbon sequestration [15]. In addition to wetland delineation, which maps the distribution, size and shape of wetlands, regional or catchment scale wetland classification maps describing wetland types are, therefore, needed to monitor and evaluate the ecological function for adaptive management [16,17]. Vegetation formations are often the foundation for wetland classification because they are relatively persistent in the hydrologically dynamic wetland environment [18], and they integrate past and current hydrological conditions [19], soil moisture, and fertility [20]. Most wetland classification systems are essentially a description of wetland vegetation types [21].

Vegetation mapping is undertaken using a variety of methods depending on the requirements of the purpose. The most demanding level of vegetation mapping requires intensive field work, including taxonomical information, and the visual estimation of percentage cover for each species [5]. This approach is time-consuming, labour intensive, and costly; thus, it is only practical across relatively small areas [22]. It can be usefully supplemented by the manual interpretation of high-resolution aerial photographs (e.g., in Reference [23]); however, the costs associated with accessing repeat photography can limit opportunities for a time series analysis. The manual interpretation of aerial photographs also relies heavily on the knowledge of the photo interpreter on vegetation community identification [24]. While high spatial-resolution aerial photography is able to provide valuable spatial information and to delineate a target boundary, photographic analyses cannot be semiautomated to discriminate the target due to the coarse spectral-resolution [25].

Some recent vegetation mapping efforts have been focused on (semi-) automating digital analysis images acquired from satellite remote sensing platforms including Landsat (mainly TM and ETM+), SPOT, MODIS, NOAA-AVHRR, IKONOS, and QuickBird [5,26]. Among these efforts, image fusion and statistical learning have proven to be efficient tools in wetland vegetation mapping. The former is an algorithm to combine relevant information from two or more images (normally from different sources) into a single image [26–28]. The latter is a data mining technique used to discover knowledge from large amounts of data [29–31].

Topography and landscape position play key roles in wetland formation [32,33]. LiDAR data can provide detailed and accurate ground elevation information for identifying topographic features and, in some cases, for indicating the boundary of wetlands that would otherwise be undetected [32]. In wetland-rich areas such as floodplains where the complex nesting of features may cause mapping confusion and inaccuracy [33], geomorphological variable-derived fine-scale elevation data can be particularly helpful [34,35].

In this study, we used stochastic gradient boosting machines (GBM, a machine learning technique) to predict vegetation communities in the Barwon-Darling, a large semiarid floodplain wetland complex in Australia. Our overall goal in this study was to explore the capacity and performance of image fusion and statistical learning in discriminating wetland vegetation types and to demonstrate to wetland ecologists the accessibility of remote sensing for the monitoring and assessment of floodplain vegetation. We have two specific objectives:

1. To investigate the discrimination limits of the predictive models through evaluating the performance of predictive models at three vegetation formation levels. Our approach involved building predictive models for three vegetation classification levels: at a broad functional group level (two classes: wetland vs. non-wetlands), an intermediate level (four classes) which further divides wetlands into vegetation structural groups, and a more detailed level involving vegetation floristics (dominant species—nine classes).

2. To evaluate the predictive power of various predictor variables. In wetland ecosystems, vegetation distribution is strongly affected by the moisture gradient, which is often highly correlated with micro-topography [36,37]. The combined use of LiDAR-based geomorphological data and Landsat data may present an opportunity for refined vegetation mapping. Our study, therefore, assesses the individual and combined contributions of geomorphological variables to classification accuracy by building models with three combinations of predictor

variables: full models with all Landsat and LiDAR DEM-based geomorphological variables; the geomorphological models with only LiDAR DEM-based geomorphological variables; and the Landsat models fitted with only Landsat variables.

## 2. Materials and Methods

### 2.1. Study Site

The study area covers more than 338,800 ha of floodplain in the Barwon-Darling River system located in the northwest New South Wales (NSW) (Figure 1). The area has a semiarid climate with low rainfall, hot summers, and cold winters. The average annual rainfall (1998–2016) is 306.2 mm. The mean maximum summer and mean minimum winter temperatures (1998–2016) are 37.2 °C and 4.2 °C, respectively [38].

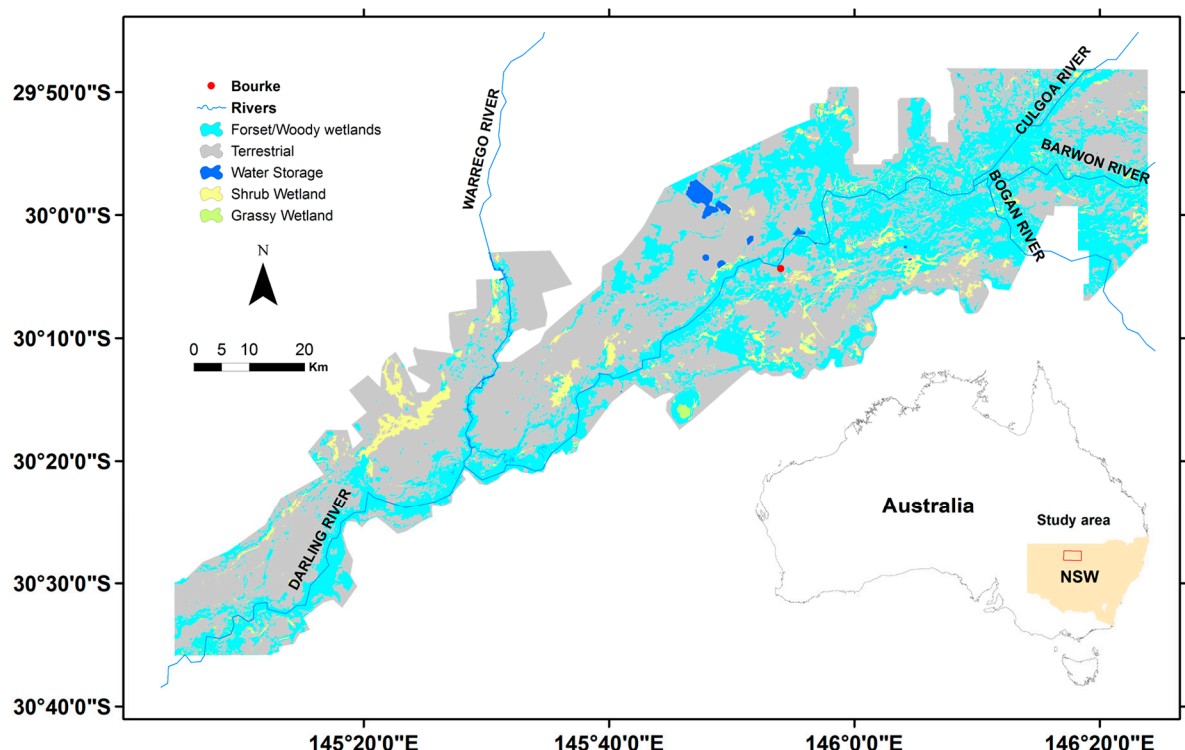

**Figure 1.** The vegetation communities in the Barwon-Darling and Condamine-Balonne floodplain systems of New South Wales (NSW), Australia: The classes were based on community structure. The inset shows the location of the study site.

The floodplain consists of a wide range of floodplain forests and woodlands, including river red gum (RRG) (*Eucalyptus camaldulensis*), black box (*E. largiflorens*), coolabah (*E. coolabah*), and river cooba (*Acacia stenophylla*). There are also areas of wet meadows including species such as spikerush (*Eleocharis* spp.), nardoo (*Marsilea* spp.), and Marshwort (*Nymphoides* spp.) scattered across the lower depressions. The native vegetation communities can be broadly separated into flood-dependent and non-flood-dependent. Flood-dependent vegetation requires periodic inundation to regenerate and survive, whereas non-flood-dependent systems rely on local rainfall and generally sit at slightly elevated terrains off the floodplain. Flood-dependent vegetation includes the riparian forests of River Red Gum (*Eucalyptus camaldulensis*) that fringe the major rivers (including the Darling, Bokhara, and Culgoa) and the extensive floodplain woodlands dominated by Coolibah (*E. coolabah*) and Black Box (*E. largiflorens*). We include these flood-dependent forests and woodlands into our wetland functional grouping. Along the ephemeral channels and within low lying depressions, additional various wetland

communities occur, including large semipermanent wetlands and inland lakes and smaller ephemeral wetlands dominated by *Eleocharis* spp., *Marsilea* spp., and *Nymphoides* spp. The shrublands dominated by Lignum (*Duma florulenta*) and Nitre Goosefoot (*Chenopodium nitrariaceum*) are commonly associated with minor channels and low-lying areas throughout the region.

## 2.2. Wetland Vegetation Map

We used vegetation maps produced by References [39–41]. The mapping process for these studies was guided by on-screen aerial photographic interpretation, the incorporation of existing map data, and an extensive program of field reconnaissance. The combined vegetation map has 26 PCTs (plant community types). Croplands and man-made features such as dams were excluded in this study. The PCT classification was developed by New South Wales Office of Environment and Heritage [42] to establish an unambiguous master community-level scheme for use in vegetation mapping programs, in BioMetric-based regulatory decisions, and as a standard typology for other planning and data gathering programs.

To test the discrimination power of statistical learning, we grouped the 26 PCTs into three hierarchical levels of vegetation classes for modelling (Table 1) and then compared the performance of the models at the three levels:

(1) Level one contains the two broadest functional groups: flood-dependent (wetland) and non-flood-dependent terrestrial vegetation communities. PCTs were assessed as either flood-dependent (wetland) or terrestrial based on knowledge of the wetland plant indicator status of dominant plant species. The wetland indicator plant species list is from derived, descriptive information recorded in the NSW Flora Online [43].

(2) Level two has three classes which divide the flood-dependent group (wetlands) into three broad structural groups: woodland, shrubland, and grassy/herbaceous wetlands. We grouped the PCTs into these structural groups based on the dominant life form of the tallest plant layer [44].

(3) Level three comprises nine vegetation classes, formed by grouping PCTs per dominant species (floristics), and the conceptual understanding of the landscape position and water requirements of these communities described in the NSW Vegetation Information System Classification database [42].

**Table 1.** The types of the Barwon-Darling used in this study.

| Level 1 | Level 2 | Level 3 | Area (ha) | Sampled Points |
|---|---|---|---|---|
| L1: Wetlands | L11: Woody wetlands | L111: River red gum forest | 8671 | 2560 |
| | | L112: Black box woodland | 14,082 | 4096 |
| | | L113: Coolibah woodland | 179,879 | 19,465 |
| | | L114: Other woodland | 16,132 | 1750 |
| | L12: Shrub dominated wetlands | L121: Lignum shrubland | 24,852 | 4601 |
| | | L122: Nitre Goosefoot shrubland | 326 | 106 |
| | | L123: Other Shrublands | 411 | 233 |
| | L13: Grassy wetlands | L131: Inland Floodplain Swamp | 2648 | 820 |
| L2: Terrestrial | L21: Terrestrial | L211: Terrestrial | 291,740 | 23,374 |

Since we focused on wetland vegetation communities, the non-flood-dependent vegetation, including salt bush, mixed chenopod shrubland, and Belah Woodland, was not separated further. With the vegetation map, we extracted 57,005 stratified (in terms of vegetation types, Table 1) random points using the Create Random Points tool in ArcGIS (ArcGIS 10.4, ESRI, Redlands, CA, USA) for the vegetation community type modelling.

*2.3. Inputs of GBM Classifier*

2.3.1. Geomorphological Variables

A detailed topographic survey was conducted for the study area by Geoscience Australia in 2014 using airborne Light Detection and Ranging (LiDAR). A 1-m DEM (digital elevation model) with a vertical accuracy of 0.15 m was built from the LiDAR 3-D point cloud model [45]. To reduce the computation time, using bilinear interpolation, we resampled the 1-m DEM to a 5-m DEM following References [46], which found that a resolution of 5 m for LiDAR surface models best balances the classification performance and computational challenges posed by large land-area assessments. From the 5-m DEM, we derived a range of primary and secondary topographic metrics using the GradientMetrics tool (available from https://github.com/jeffreyevans/GradientMetrics) developed for ArcGIS. The primary variables include landform and COV, and the secondary variables include CTI:

1.  CTI: Compound Topographic Index, a steady state wetness index [47];
2.  Landform: surface curvature (concavity/convexity) index [48];
3.  COV: the coefficient of variation of elevation within a 100 m × 100 m window; and
4.  LDFG: local deviation from a 100 m × 100 m global window.

Since flow is a key driver for floodplain vegetation distribution [49], we computed the cost distance: the Euclidean distance (m) to streams and rivers weighted by elevation. These variables are related to landscape shape and position and are the main determinants of inundation [47,49]. We also computed the same geomorphological variables from the 1-arc second Hydrologically Enforced DEM (from Geoscience Australia: www.ga.gov.au/topographic-mapping/digital-elevation-data.html), referred to as the 30-m geomorphological variables. The actual elevation, which was highly associated with LDFG (Pearson's $r$ of $-0.99$), was not included in the modelling. The paired correlation coefficients (Pearson's $r$) for the selected predictors are low (Pearson's $r$ less than 0.50).

2.3.2. Variables derived from Landsat TM and ETM+ Images

1.  Inundation Counts

An inundation raster was developed using Landsat images spanning the period of 2000 to 2016. We selected images with lower percentages of cloud cover (cloud cover less than 25%) and with greater areas of inundation (i.e., we selected images from wetter periods) using a model built in ArcGIS. Each image was processed to a binary (water/non-water) raster using the water index developed by Reference [50]. The binary rasters were then summed to produce the inundation count raster.

2.  Fractional Cover Polar Statistics

While a single Landsat image can be insufficient for discriminating vegetation types in wetland environments due to seasonal variations in condition [51], studies have shown that a multi-temporal analysis can achieve a relatively accurate classification [52,53]. A time series of the seasonal fractional cover developed from Landsat TM and ETM+ images are available from AusCover (http://www.auscover.org.au). The individual Landsat images were converted to standardised surface reflectance measurements [54] and then combined to produce seasonal composite images, providing four composite images per year. The fractional cover model is applied to the Landsat imagery to predict the fractions of bare ground, green vegetation and nongreen vegetation, and a model fitting error [55].

The time series (1987–2013) of seasonal fractional cover images were used to develop metrics that characterised the variation in green and nongreen vegetation. First, an individual fractional cover image, which encloses the proportion of green vegetation, nongreen vegetation, and bare soil within each pixel (55), was converted into a polar coordinate system, and the fraction of green and nongreen vegetation was expressed as the distance from the origin to the reference point and the polar angle

(θ) between the projected line from the origin to the reference point and the positive horizontal axis (green vegetation) (Figure 2). Second, the time series pixel values of the distance, angle, and hop length (defined as the distance between two successive points in the polar space) were used to produce a range of statistics. We selected the following five after a primary assessment of the applicability using simple linear regression.

a.      The polar angle median (V1)
b.      The vector distance median (V2)
c.      The vector distance maximum (V3)
d.      The hop length median (V4)
e.      The skewness in the vector distance based on the statistical distribution of the distance values at a pixel (V5)

We checked the correlation between the predictor variables using the Pearson correlation coefficient. A highly correlated variable was excluded, and the correlations among the remaining predictors are low to moderate (paired Pearson's $|r|$ ranges from 0.04 between Landform and LDFG to 0.72 between V3 and V4).

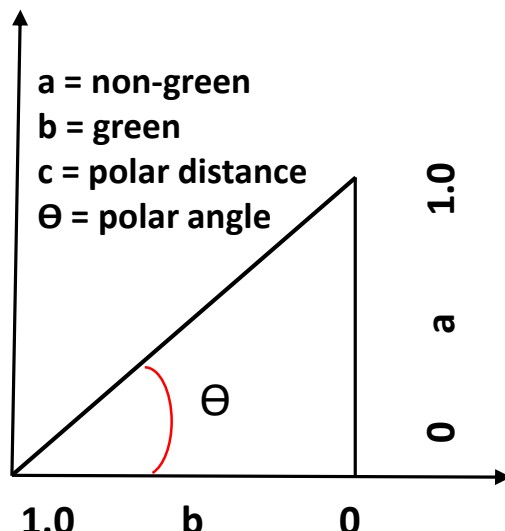

**Figure 2.** The projection of fractional data into a polar coordinate system.

*2.4. Stochastic Gradient Boosting Machines*

We used GBM methodology implemented in the R package "gbm" [56] to build the predictive models for the vegetation community. We built three sets of models, one for each level of wetland class (Table 1). In the GBM algorithm, regression trees are sequentially fitted to residuals to minimize the loss function, after which a new tree is added to the model. Many weak regression trees are combined (by restricting tree depth) in a successive fashion (i.e., boosting) to optimize the predictive performance [57,58]. To avoid overfitting, the contribution of each added tree to the previous iteration's predicted value is reduced by the learning rate λ.

The dataset was split into training (25% or 14,259 samples) and testing (75% or 42,746 samples) subsets using stratified random sampling. We used a smaller portion of data for model training to restrain the computing time. Moreover, we deliberately used most samples as a testing dataset to examine the stability of the trained models. With the training dataset, a 5-fold repeated (5 times) cross-validation method was used for the model tuning.

GBM has four hyperparameters to optimize: number of trees, learning rate, tree complexity, and minimum number of observations. In searching for the best model, we fixed the minimum number of observations to be 10 and let the other three vary: tree depth could be 1, 3, 5, or 7; the learning rate

could be 0.01, 0.03, . . . , or 0.1; and the number of trees could be 100, 150, . . . , or 500. The GBM models were fitted to all combinations of the three hyper-parameters, and the optimal model was selected on the basis of overall accuracy (OAA).

## 2.5. Accuracy Assessment and Model Comparison

We report four indicators to compare the performance of the models: OAA, balanced accuracy (BA), Kappa coefficient of agreement, and AUC (Area Under the Receiver Operating Characteristic curve, or ROC Curve). As the dataset is highly unbalanced (Table 1), BA, which is calculated as the mean of Sensitivity and Specificity and conceptually defined as the average accuracy obtained on all classes, might provide a better indicator of the model performance.

To assess the value of the Landsat variables (i.e., inundation prevalence value and polar statistics of fractional cover time series), three GBMs were fitted using the above procedures: M1 with all predictor variables; M2 fitted with only topographic variables; and M3 fitted with only Landsat variables. In addition, to evaluate the freely available 1-s SRTM (Shuttle Radar Topography Mission) DEM in vegetation community classification, we built predictive models with the 30-m geomorphological variables and Landsat variables (M4) and the 30-m geomorphological variables only (M5). All predictor variables were normalized before model fitting.

For each of the five models, the variable influence for the decision tree ensembles was computed based on the decision trees influences [59] using the method proposed by Reference [60]. To facilitate a comparison, we scaled the total influence to be 100 and calculated the relative importance of every variable.

The performance of the five models was evaluated in two ways. First, a permutation t-test implemented in the R package "perm" [61] was conducted for the final train model performance metrics (i.e., OAA and Kappa). Permutation tests are a class of widely applicable nonparametric tests. They use random shuffles of the data to get the correct distribution of a test statistic under a null hypothesis [62]. Second, we compared and ranked the performance of the five models using the testing sub-dataset. McNemar's test [63], which is nonparametric and based on the model error matrices, was used to test the significance in the difference of model performance. The difference in accuracy for a pair of classifications is considered as being statistically significant at the 95% confidence level if the z-score is larger than 1.96 in the McNemar test. This test has previously been used to compare the accuracies of two or more classifications [64] and is known for its efficiency in comparing classification accuracies [65]. The difference was also tested using a paired permutation t-test based on 100 resamples of the final models for each of three performance criteria.

## 3. Results

### 3.1. Model Performance

#### 3.1.1. Level 1 Models

The training model performance metrics for the Level 1 models are shown in Figure 3. The performances of Level 1 models M1 and M4 were superior to all other models. The model rank from highest to lowest values of Kappa and OAA was M1, M4, M3, M2, and M5 (Figure 3). The difference in accuracy between M1 and M4 was insignificant through a comparison of the box and whiskers in Figure 3. This was confirmed by the paired permutation t-tests on OOA between M1 and M4 (mean $p = 0.822$ with 99% confidence interval of 0.741 and 0.903 based on 999 Monte Carlo replications). The difference between all other pairs was significant ($p$ ranges from 0.000 to 0.002). The model evaluation based on Kappa showed similar results.

The predictive power of the models was evaluated using the testing dataset, and they ranged from fair to substantial (Table 2). Model M1, which used both Landsat based variables and geomorphological metrics computed from the 5-m LiDAR DEM achieved the highest overall accuracy of 83.73%. Model

M4, which used the 30-m geomorphological variables and Landsat variables ranked second (OAA 83.27%). There was no significant difference between these two models, and the accuracy was considered substantial based on Kappa [66]. Model M5, which used only the 30-m geomorphological predictors, had the worst performance with an overall accuracy of 70.72%. With AUC greater than 0.70, all five models but M5 were assessed as acceptable for conservation purposes [67].

Using the overall correctly and erroneously predicted cases as a two-dimensional contingency table, we examined the difference in model performance using the nonparametric McNemar's test. While Model M1 was slightly better than Model M4, the difference was not significant (McNemar's chi-squared = 2.429, df = 1, $p$ = 0.119). The other paired tests were all significant ($p < 0.001$).

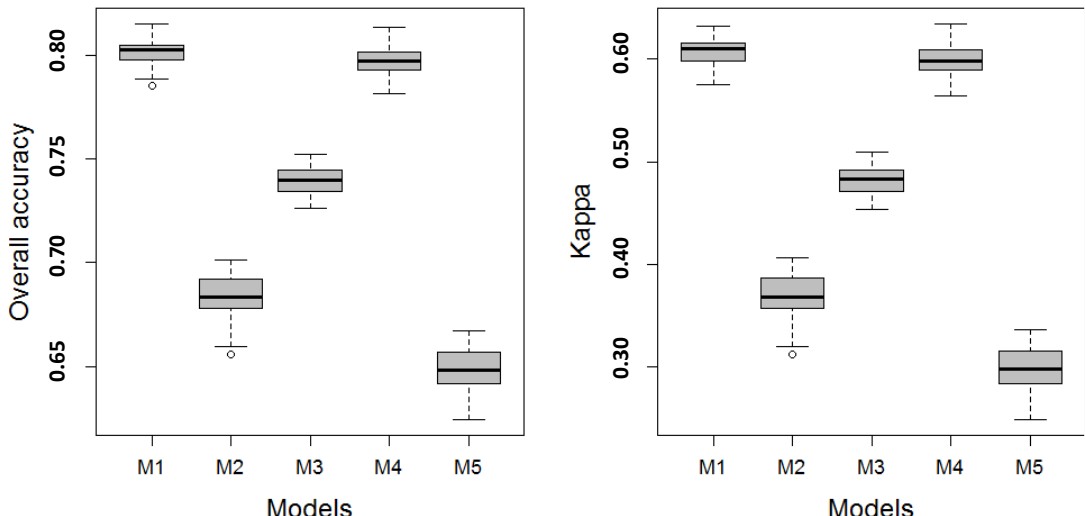

**Figure 3.** The metrics shown as boxplots of overall accuracy (OAA) and Kappa for Level 1 vegetation classes were based on a resampling of the final model ensemble (n = 100). The boxplots convey the median (dark line), the lower (25%) and upper (75%) quartiles (boxes), and the minimum and maximum (whiskers). The outliers are indicated by circles.

**Table 2.** The prediction accuracy of Level 1 models using the testing dataset, which is independent to the training dataset.

| Model | L1 | | L2 | | BA [3] | OAA [4] | Kappa | AUC [5] | Verdict [6] |
|---|---|---|---|---|---|---|---|---|---|
| | UA [1] | PA [2] | UA | PA | | | | | |
| M1 | 83.54 | 88.63 | 83.96 | 77.30 | 82.97 | 83.73 | 0.64 | 0.80 | Substantial |
| M2 | 75.37 | 80.75 | 72.42 | 65.70 | 73.22 | 74.20 | 0.44 | 0.70 | Moderate |
| M3 | 78.92 | 84.33 | 77.64 | 70.72 | 77.55 | 78.41 | 0.54 | 0.75 | Moderate |
| M4 | 82.82 | 88.81 | 83.95 | 76.07 | 82.44 | 83.27 | 0.63 | 0.80 | Substantial |
| M5 | 68.21 | 61.16 | 72.32 | 78.07 | 69.64 | 70.72 | 0.37 | 0.67 | Fair |

[1] UA = User's Accuracy; [2] PA = Producer's Accuracy; [3] Balanced accuracy = (Sensitivity + Specificity)/2; [4] Overall accuracy = the proportion of the total number of predictions that were correct × 100; [5] Area Under the ROC Curve; [6] Based on Kappa according to Landis & Koch (1977): 0–0.20, slight; 0.21–0.40, fair; 0.41–0.60, moderate; 0.61–0.80, substantial; and 0.81–1, almost perfect.

### 3.1.2. Level 2 Models

The training model performance metrics for the Level 2 models are shown in Figure 4. Like the Level 1 models, the performances of Level 2 models M1 and M4 were superior to all other models. The model ranking from highest to lowest values of Kappa and OAA was M1, M4, M2, M3, and M5 (Figure 4). The difference in accuracy between M1 and M4 was insignificant by visually comparing the box and whiskers in Figure 3. The nonsignificant difference in model performance was confirmed by the paired permutation t-tests on OOA and Kappa between M1 and M4 ($p$ = 0.336 and 0.246 for

the OOA and Kappa comparisons, respectively). The difference between all other model pairs was significant ($p < 0.01$).

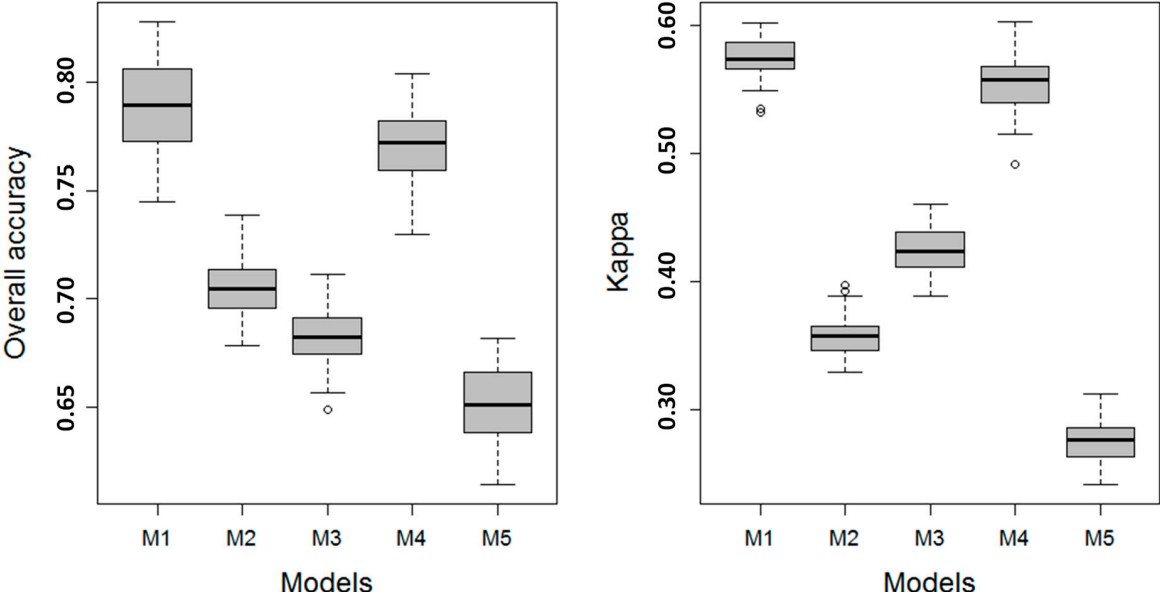

**Figure 4.** The metrics shown as boxplots of overall accuracy (OAA) and Kappa for Level 2 vegetation classes were based on a resampling of the final model ensemble (n = 100). The boxplots convey the median (dark line), the lower (25%) and upper (75%) quartiles (boxes), and the minimum and maximum (whiskers). The outliers are indicated by circles.

The ranking of the models based on predictive power followed the same order as that suggested by the training metrics (Table 3). Model M1 achieved the highest overall accuracy of 79.57%, which was noticeably but not significantly different to the accuracy of Model M4 at 78.92% (Figure 3). Using the overall rightly and wrongly predicted cases, the paired McNemar's test confirmed that there was no significant difference between M1 and M4 (McNemar's chi-squared = 1.768, df = 1, $p$ = 0.184), but the difference was significant for all other paired tests ($p$-values ranged from 0.000 to 0.007). Model M5, again, had the lowest overall accuracy at 64.00%.

**Table 3.** The prediction accuracy of Level 2 models using the testing dataset, which is independent to the training dataset.

| Model | L11 | | | L12 | | | L13 | | | L21 | | | OAA | Kappa | AUC | Verdict |
|---|---|---|---|---|---|---|---|---|---|---|---|---|---|---|---|---|
| | UA | PA | BA | UA | PA | BA | UA | PA | BA | UA | PA | BA | | | | |
| M1 | 82.80 | 83.89 | 83.63 | 69.39 | 72.69 | 84.03 | 77.36 | 70.66 | 85.31 | 78.95 | 76.44 | 82.06 | 79.57 | 0.75 | 76.12 | Substantial |
| M2 | 70.93 | 75.78 | 73.06 | 58.75 | 55.31 | 74.85 | 69.23 | 66.88 | 83.39 | 66.93 | 62.37 | 71.86 | 68.04 | 0.55 | 64.18 | Moderate |
| M3 | 78.32 | 77.48 | 78.50 | 52.45 | 61.70 | 76.78 | 55.58 | 34.43 | 67.17 | 69.94 | 67.17 | 74.85 | 69.30 | 0.62 | 67.97 | Moderate |
| M4 | 82.43 | 83.55 | 83.28 | 68.38 | 72.09 | 83.64 | 92.21 | 57.45 | 78.75 | 77.90 | 75.58 | 81.31 | 78.92 | 0.74 | 75.83 | Substantial |
| M5 | 67.93 | 71.83 | 69.72 | 52.99 | 53.39 | 73.26 | 70.82 | 54.81 | 77.38 | 62.15 | 57.60 | 68.19 | 64.00 | 0.49 | 61.06 | Fair |

See Table 1 for the vegetation classes and Table 2 for explanations of the performance metrics.

Based on Kappa coefficients, models M1 and M4 were considered substantial, M2 and M3 were moderate, and M5 was fair (Table 3). With AUC less than 0.70, Models M2, M3, and M5 were considered unacceptable for conservation purposes [67]. With balanced accuracies greater than or close to 80% for all classes, M1 and M4 achieved a satisfactory vegetation classification (Table 3).

For the best performing model, M1, we compared the discriminative power of the model for each of the vegetation classes using the paired McNemar's test. While the model had a comparable predictive capacity for flood-dependent woody wetlands and non-flood-dependent terrestrial systems (McNemar's chi-squared = 1.792, df = 1, $p$ = 0.181), the discriminative power was significantly lower for

grassy and shrub dominated wetlands (*p* < 0.001). For the least accurately classified flood-dependent shrub-dominated wetlands, the confusion was mainly due to a difficulty in separating them from terrestrial communities and flood-dependent woodlands (i.e., there was less confusion between grasslands and shrublands). For grassy wetlands, however, the misclassification was mainly due to the erroneous allocation of forest/woodland wetlands into the target classes.

In order to test the difference in discriminative power between Level and Level 2 paired models, we again used McNemar's test. The results indicated that the models for level two classification had significantly lower discriminative power (*p* < 0.001) for each pair, except for M4, which had a comparable performance for both levels (McNemar's chi-squared = 1.455, df = 1, *p* = 0.228).

### 3.1.3. Level 3 Models

The performance metrics for the Level 3 training models are shown in Figure 5. The performance ranking was M1, M4, M3, M2, and M5 (Figure 5). For M1 and M4, which the performances of were comparable for the Level 1 and Level 2 classes, the performances were significantly different for the Level 3 classes (permutation *t*-tests, *p* = 0.01 and 0.004 for Kappa coefficient and OAA, respectively). Nevertheless, the prediction performance of the two models was similar according to McNemar's test using the total correctly and wrongly allocated cases as contingency table (McNemar's chi-squared = 2.429, df = 1, *p*-value = 0.119). The permutation t-tests indicated the performance differences between the remaining models was significant for all paired comparisons.

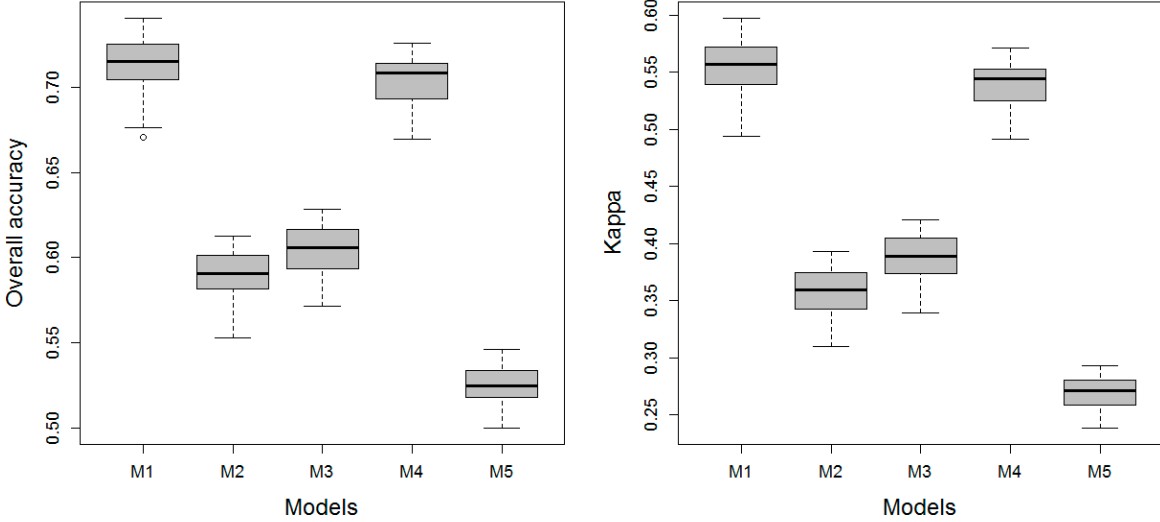

**Figure 5.** The metrics shown as boxplots of overall accuracy (OAA) and Kappa for Level 3 vegetation classes were based on a resampling of the final model ensemble (n = 100). The boxplots convey the median (dark line), the lower (25%) and upper (75%) quartiles (boxes), and the minimum and maximum (whiskers). The outliers are indicated by circles.

The Kappa coefficients showed that M1 and M4 had a substantial predictive power, and the performance of M2 and M3 was moderate, whereas M5 was considered fair (Table 4). Both M1 and M4 were considered acceptable based on AUC [67] (Table 4). These performance assessments were the same as the results reported for the Level 2 models, indicating a limited debilitation of the model performance when increasing the number of classes from four to nine.

There are nine wetland types in the Level 3 classification. Although assessed as having a substantial discriminative capacity, the top two models (M1 and M4) achieved satisfactory results for only four of the seven classes based on the producer's and user's accuracy, namely River Red Gum forests, lignum dominated shrublands, inland floodplain swamps, and terrestrial class (Table 4). The models were particularly efficient for River Red Gum forests with over 75% user's and producer's accuracy (Table 4). However, the models were ineffective in distinguishing Black Box woodlands,

Coolibah woodland, and Nitre Goosefoot-dominated shrublands from other classes. For Black Box woodlands, apart from being misclassified as non-flood-dependent uplands, the confusion within the same functional group was the largest source of error. Out of the 753 cases of Black Box woodland, 304 and 324 were classed as other woodlands by M1 and M4, respectively. For the two types of shrub-dominated wetlands, however, the confusion mainly came from forest/woodland (both Black Box woodlands and other forest/woodland types) and the terrestrial (non-wetland) class.

The paired McNemar's tests indicated that the models for Level three classification had a significantly lower discriminative power than those for Level two ($p < 0.001$).

**Table 4.** A summary of the Level 3 model prediction power, using testing dataset.

| Classes | | M1 | M2 | M3 | M4 | M5 |
|---|---|---|---|---|---|---|
| L111 | UA | 79.23 | 72.58 | 68.25 | 78.77 | 56 |
| | PA | 75.16 | 72.86 | 62.43 | 70.79 | 56.22 |
| | BA | 87.24 | 85.96 | 80.73 | 85.06 | 77.34 |
| L112 | UA | 58.18 | 50.23 | 60.15 | 71.98 | 42.14 |
| | PA | 38.84 | 33.00 | 21.44 | 34.86 | 35.92 |
| | BA | 69.14 | 66.17 | 60.62 | 67.27 | 67.48 |
| L113 | UA | 63.73 | 52.47 | 57.45 | 62.84 | 48.7 |
| | PA | 65.95 | 57.29 | 59.21 | 66.11 | 51.91 |
| | BA | 66.87 | 57.44 | 61.01 | 66.47 | 53.47 |
| L114 | UA | 51.88 | 43.93 | 53.85 | 65.68 | 35.84 |
| | PA | 32.54 | 26.7 | 15.14 | 28.56 | 29.62 |
| | BA | 62.84 | 59.87 | 54.32 | 60.97 | 61.18 |
| L121 | UA | 67.84 | 71.08 | 54.81 | 91.5 | 67.52 |
| | PA | 73.58 | 69.74 | 33.89 | 57.29 | 55.02 |
| | BA | 84.39 | 75.68 | 76.89 | 84.19 | 73.7 |
| L122 | UA | 56.33 | 39.71 | 32.73 | 51.37 | 32.25 |
| | PA | 39.38 | 25.46 | 16.66 | 35.12 | 19.21 |
| | BA | 63.86 | 56.85 | 52.45 | 61.71 | 53.71 |
| L123 | UA | 52.15 | 35.53 | 28.55 | 47.19 | 28.07 |
| | PA | 35.20 | 21.28 | 12.48 | 30.94 | 15.03 |
| | BA | 59.68 | 52.67 | 48.27 | 57.53 | 49.53 |
| L131 | UA | 78.21 | 58.84 | 51.61 | 67.64 | 51.38 |
| | PA | 72.01 | 56.94 | 61.67 | 73.24 | 54.44 |
| | BA | 85.91 | 84.75 | 66.82 | 78.59 | 77.39 |
| L211 | UA | 78.67 | 67.06 | 69.53 | 77.72 | 61.92 |
| | PA | 76.06 | 63.63 | 66.82 | 75.28 | 58.1 |
| | BA | 81.77 | 72.32 | 74.53 | 81.08 | 68.2 |
| OAA | | 76.95 | 66.15 | 67.95 | 76.36 | 60.8 |
| Kappa | | 0.57 | 0.4 | 0.43 | 0.56 | 0.32 |
| AUC | | 0.65 | 0.6 | 0.56 | 0.64 | 0.53 |
| Verdict | | Substantial | Moderate | Moderate | Substantial | Fair |

See Table 1 for the vegetation classes and Table 2 for explanations of the performance metrics.

### 3.2. Relative Variable Influence

Although the relative importance of variables varied among models for different vegetation levels, the changes were marginal and the ranks of predictor importance generally remained consistent. In addition, the rank of variables in simpler models (i.e., with less predictors) largely stayed the same in the more complicated models. To illustrate, we determined the relative variable influence for the two more complicated models for Level 2 wetland vegetation classes (Figure 6). Generally, both topographic variables and multispectral indices computed from satellite imagery were important in a

successful wetland vegetation classification (the relative contribution of topographic predictors was 55.34% and 47.45% for M1 and M4, respectively (Figure 6).

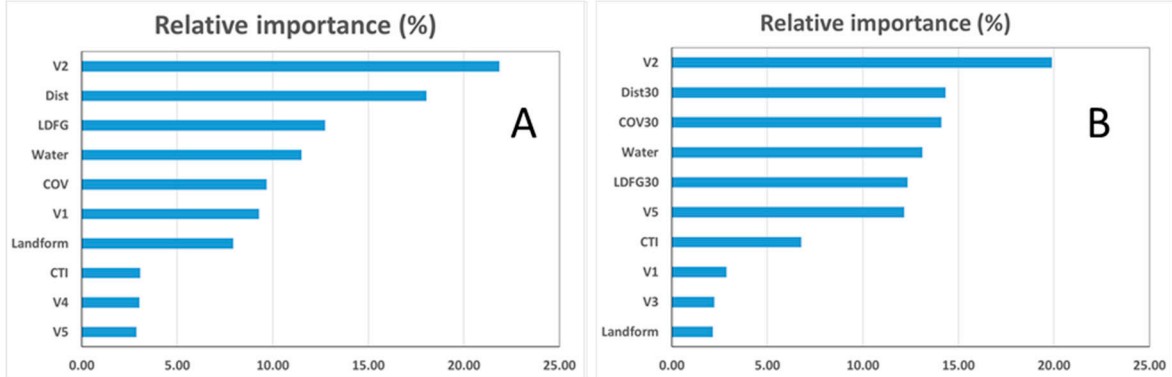

**Figure 6.** The relative influence of variables in Model M1 (**A**) and Model M4 (**B**) for Level 2 wetland types: The importance scores are scaled so that the total is 100.

For both models, V2, a measure of the long-term mix of green and dead components within a grid, was the most important predictor for floodplain wetland classification (21.86% and 19.90 % for M1 and M4 respectively, Figure 5). The cost distance (calculated as the Euclidean distance to streams and rivers weighted by elevation), a hydro-geomorphic variable that quantifies the flood potential of a location, was the second most influential predictor. LDFG, the topographic metric describing the local deviation from global in elevation, ranked the third as an important predictor for M1 (a similar topographic variable, COV, the coefficient of variation in elevation within a 100 m × 100 m, was the third for M4). Other important predictors (in order of relative importance) included COV, V3, Landform, and V5. For M4, COV, the coefficient of variation in elevation within a 100 m × 100 m window, was the second important predictor. Other important predictors included the inundation counts (ranked 4th in both models), V1, CTI, and V5 (Figure 5).

## 4. Discussion

### 4.1. The Discriminative Limits of Predictive Models Using Landsat and DEM-Derived Predictors

We evaluated the final models using a much larger (5.56 times more) independent samples. The performance criteria (i.e., Kappa, OAA, BA, and AUC) calculated from the testing results were comparable to those of the training models, suggesting there was little concern of overfitting. The stable predictive performance was achieved as a result of the multiple 10-fold cross-validation model tuning process.

As our objective is to investigate the discriminative limit of statistical learning, the discussion was focused on the best models. Although the prediction of the models was considered as acceptable based on the performance metrics (i.e., Kappa and AUC, [66,67], the overall accuracy of our models was less than that reported in previous studies using a similar approach. For example, using a phenological metric extracted from long-term Landsat images [68] was able to discriminate a wetland forest from an upland forest with high accuracy (88.5%). However, the performance declined significantly when the number of vegetation types increased to 3 (82.2%). Similarly, Reference [53] mapped the land cover of Chile using the time series of Landsat images and SRTM DEM with an overall accuracy of 80%, 73%, and 59% for the Level 1 (10 classes), 2 (30 classes), and 3 (35 classes) classification schemes. There are a number of possible reasons for the relative underperformance of our study. First, the misclassification in the original vegetation map may increase the errors as the modelling performance is also dependent on the accuracy of reference data [69]. For models at all levels, the majority of the confusion came from the misallocation of non-wetland into the targeting classes. For example,

out of the 11,718 non-wetland testing samples, the Level one model incorrectly classified 1551, 664, and 516 as flood dependent woodlands, flood-dependent shrublands, and semipermanent grassy wetlands, respectively. In the semiarid Barwon-Darling catchment, many wetlands, especially the riparian forests and grass-dominated wetlands, are small or narrow, formed in a variety of landforms (Figure 1). The hydrology in these wetlands is affected by small changes in elevation [70], complicating their identification. In many cases, the vegetation communities of the drier wetlands (such as lignum dominated swamps) are not dramatically different at wetland boundaries from the adjacent uplands, making wetland demarcation even more difficult. Second, a class imbalance can adversely impact the performance of a predictive model [71]. In our study, more than 50% of the data points were non-wetland resulting from the random stratified sampling as a large portion (>50%) of the floodplain was mapped as arid shrublands and semiarid woodlands, which was classified as terrestrial (Table 1). Although resampling methods (such as up-sampling or down-sampling) can quickly deal with the problem of unbalanced samples, they give more importance to the data in the majority class and neglect potentially important data in the minority class, thus limiting the effectiveness and repeatability of classification [72]. Class imbalance remains an issue in machine learning research [71].

As expected, the overall performance of models decreased with increasing wetland classes. The functional group level model (four classes, Table 1) was considered as substantial based on the Kappa coefficient (Kappa = 0.76 and 0.75 for training and testing, respectively) achieved an OAA of 79.57%. Although the performance of the Level 3 model (with Nine classes) was also high with an OAA of 76.57%, the discriminative powers for the individual classes varied and was considered as lower for black box woodlands, other woodlands, Nitre Goosefoot Shrublands, and other shrublands based on the user's and producer's accuracy (Table 4). The difficulty arises because of the similarity in the spectral reflectance properties of the vegetation communities with a comparable structure and physiognomy [5]. The temporal changes in the community composition could also contribute to the misclassification. Seed banks in a semiarid Australian floodplain are very persistent and capable to respond quickly to the unpredictable rainfall events [18,73]. During an over-three-decades-long study period, the vegetation community in some areas may undergo fundamental changes and transit from one PCT to another. For example, water couch (wetland species) can colonize chenopod shrublands (terrestrial community) rapidly and at a large spatial scale in wet conditions [74] and authors' field observation]. Despite the decrease in OAA with the increase of wetland classes, the models at all levels were highly effective to distinguish terrestrial uplands, and the paired McNemar's tests suggested that there was no significant difference in the performance of the models in terms of isolating the terrestrial vegetation from wetlands (balanced accuracy was 86.09%, 82.06%, and 81.77% for the Level 1, 2, and 3 models). Also, grass-dominated swamps were also discriminated adequately, and the prediction power did not decline significantly with increasing wetland classes. In addition, our approach had a high discriminative power in isolating the Riparian River Red Gum Forest from other types of wetlands (producer's and user's accuracies were 79.23% and 75.16%, respectively, Table 4), although they only occupied a small portion of the floodplain (1.61%, Table 1).

Hyperspectral remote sensing imagery (e.g., HyMap, CASI, and Hyperion) provide a promising tool to distinguish these underperformed wetland types [5,75]. For example, combining aerial photography, hyperspectral imagery, and LiDAR [26] produced a detailed vegetation map (11 classes) with an overall accuracy of 91.1% and Kappa value of 0.89 in a portion of the central Florida Everglades. However, substantial challenges to using the hyperspectral data, such as a high cost, a large data volume, and the complexity of image processing (therefore, long processing time and specialized software), have restricted their widespread application. For these reasons, hyperspectral data are not used operationally for wetland mapping other than for particularly difficult problems, such as the discrimination of vegetation species (e.g., types of mangroves, [76]).

*4.2. The Importance of Integrating Geomorphological Variables in Floodplain Vegetation Mapping*

As hydrological features, the distribution of wetlands (and the vegetation communities within them) is clearly in part controlled by topography and landscape position [77]. Consequently, a number of researchers investigated the value of topographic variables for wetland mapping (for examples, see References [78–81]). In agreement with these studies, our results showed that topographic metrics are indeed valuable for mapping wetland types at the three tested wetland class levels: the OAA increased 5.3%, 10.3%, and 9.0% by integrating LiDAR DEM-derived geomorphologic variables for the Level one, two, and three predictive models, respectively. Furthermore, these improvements were significant for both the training and testing results indicated by the nonparametric McNemar's test. Nevertheless, the performance of the models using geomorphological variables alone was less satisfactory, and OAA (and other performance criteria) decreased for more detailed classification. It should be noted that LiDAR generates rich data about the structure of the vegetation community, such as canopy height, which have been successfully applied in wetland vegetation mapping (e.g., in Reference [82]). In this study, we only used the DEM, and did not explore the full potential of the LiDAR data.

As with the geomorphologic variables derived the more detailed LiDAR DEM, we found that geomorphologic variables computed from SRTM DEM were also useful for enhancing the models' discrimination power. The OAA increased 4.86%, 9.62%, t and 8.41% for the Level one, two, and three models, respectively. More importantly, we found no significant difference in the overall accuracies of the models using the finer 5-m LiDAR DEM and those using the coarser 30-m SRTM DEM, albeit the collective contribution of topographic variables was much smaller for the coarser SRTM DEM. This may be due to the incomparable spatial resolution between the LiDAR DEM and the Landsat, which exerted a greater influence on the model performance. Nevertheless, References [79,83] also reported that the source of elevation data and the type of topographic derivatives used in wetland classification were not major factors. This is a significant finding considering the great coverage of the freely available SRTM DEM. For example, the 30-m SRTM DEM is available for all of Australia while only a fraction has LiDAR coverage (3.14%, [45]). However, the models using coarse geomorphological variables alone were evaluated as inadequate to distinguish wetland types at all levels based on Kappa statistics.

A variety of terrain variables have been used in wetland mapping. Although many studies have demonstrated the value of geomorphologic variables, there are some inconsistence in the assessment of the specific metrics. For example, References [79] found that the primary metric slope was important for differentiating palustrine wetlands from uplands in Yellowstone National Park, USA, and the secondary metric CTI was of comparatively little value. In contrast, CTI is the key variable for discriminating wetlands from uplands in Minnesota, USA [34]. Reference [80] found that both the primary (surface curvature) and secondary (CTI) metrics were similarly valuable for modelling wetland types. The inconsistency maybe be due to the scale of the study and the degree of landscape heterogeneity. Of the six derived geomorphological metrics, we found that LDFG, a primary metric quantifying the topographic position of the focus cell [84], was the most important one. LDFG is capable of distinguishing the subtle changes in complex landscapes [85] and, therefore, might be suitable for the floodplains, which are very flat and relatively fragmented resulting from flow regulation. Another variable, the elevation-weighted distance from stream, also had a considerable contribution to differentiating wetland types. On the other hand, we found that the commonly used secondary indices (i.e., CTI) had limited influence on the performance of the models, with a relative importance less than 5%. Additional researches should be carried out to explore the relative merit of these variables for mapping wetlands in geographic settings such as coastal regions.

## 5. Conclusions

A detailed vegetation map is needed for wetland conservation and restoration as different vegetation communities may have distinct requirements such as flow regimes. It is a continuous challenge to map the distribution of different wetland types on a regional scale, and a trade-off

between categorical details and the availability of resources to ensure broad applications is often necessary for operational mapping. A number of studies have demonstrated the value of topographic variables [34,35,81] and multitemporal satellite images [52,53] in mapping wetland at site and regional scale. In this study, we have evaluated the capacity and performance of statistical learning in discriminating wetland types using a freely available Landsat time series and geomorphological variables computed from LiDAR and STRM DEM. Our study showed that there was a discrimination limit of statistical learning with the Landsat time series and geomorphological variables in wetland mapping. The approach was clearly inadequate in distinguishing certain wetland types, which are often located at the boundaries of drier wetlands. In semiarid Australia, our results suggested that the appropriate level for floodplain wetland mapping included four classes: tree-dominated forests and woodlands, shrublands, vegetated swamps, and non-flood-dependent terrestrial communities. Our results also demonstrated that the geomorphological metrics significantly improved the accuracy of wetland classification. Furthermore, geomorphological metrics derived from the freely available coarser resolution STRM DEM were as beneficial for wetland mapping as those extracted from the finer scale LiDAR DEM. The findings enable the widespread application of our approach as both data sources are freely available globally.

**Author Contributions:** Conceptualization, L.W. and M.P.; methodology, L.W., M.P., T.D.; software, L.W., G.H.; validation, L.W.; formal analysis, L.W., M.P.; investigation, M.P., G.H., T.D., J.L., M.H. and L.W.; writing—original draft preparation, M.P. and L.W.; writing—review and editing, M.P., G.H., T.D., J.L., M.H. and L.W.; visualization, L.W.; supervision, M.H.

**Funding:** The study was partially funded by NSW OEH Wetland Inventory: Pilot Study.

**Conflicts of Interest:** The authors declare no conflict of interest.

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
