# Peer review of "Mapping Wetland Types in Semiarid Floodplains: A Statistical Learning Approach"

_remotesensing, doi:10.3390/rs11060609_

Round 1

Reviewer 1 Report

- Could you add the reference to the ArcGIS module you used in section 2.3.2.1? Which kind of source images did you use? specific bands?

Author Response

We added “ArcGIS 10.4, ESRI, Redlands, CA, USA”.

The method of Fisher, Flood and Danaher (2016) was used for automated water mapping. The water index incorporates bands 2, 3, 4, 5, and 7.  

Reviewer 2 Report

I would like to commend the authors on a well-executed study. This was one of the better manuscripts that I have been asked to review over the past few years. Below are some suggestions.

I would consider adding a brief discussion of how variable importance is calculated using gradient boosting. Since this method was used to assess the importance of your input variables, a discussion would be useful to improve the interpretability of the results.

Can you provide a brief justification for the terrain variables used? Given that there are a wide variety of terrain variables that can be derived from a DEM, it is important to not why these specific variables were chosen.

Can you elaborate further on how the training and validation samples were collected? Based on your description, my interpretation is that wall-to-wall vegetation maps were available for your study area extent. Pixels within the mapped extents were then divided into training and validation sets using stratified random sampling. Is this correct? Also, does each sample represent a 30 by 30 meter Landsat pixel?

What resampling method was used to resample the LiDAR-derived DEM data?

Would it be possible to provide the equations for the terrain derivatives used here?

For the cost distance surface, would it make more sense to use topographic slope as a measure of cost or impedance as opposed to elevation? Can you explain why elevation was used as the measure of impedance here?

Can you explain what monthly ranges are included in each of the four seasonal composites?

The transformation of the time series data to polar coordinate space needs further explanation. I have not worked with these methods, and I assume that other readers may also not be familiar with these methods. Could you provide further elaboration regarding these measures and their applications in remote sensing? Please also clarify what ranges of dates were used in the analysis.

Since balanced accuracy is used in the accuracy assessment, I would explain this measure in the 2.5 Accuracy assessment and model comparison section.

Did your study provide any insight as to what seasonal composite was most useful in this classification? I have read that spring data is commonly useful for wetland mapping. Do your results suggest this also?

Some of your boxplot figures use two decimal places for Kappa while others use one. I would suggest using two decimal places in all figures for consistency.

Would it be possible to include at least one error matric? This would be helpful for the reader to understand the sources of confusion and would be most interesting for the classification involving the largest number of classes.

Would it be possible to change the axis labels for the Figure 5 importance graphs to provide more descriptive names for the variables?

I would recommend including the following citation:

Maxwell, A.E., T.A. Warner, and M.P. Strager, 2016. Predicting palustrine wetland probability using random forest machine learning and digital elevation data-derived terrain variables, Photogrammetric Engineering & Remote Sensing, 82(6): 437-447. https://doi.org/10.1016/S0099-1112(16)82038-8.

This paper assesses the value of different terrain variables for mapping palustrine wetlands. It also notes the importance of cost distance as distance from streams weighted by slope in mapping palustrine wetlands.

I would also recommend including the following citation:

Maxwell, A.E., and T.A. Warner, 2019. Is high spatial resolution DEM data necessary for mapping palustrine wetlands?, International Journal of Remote Sensing, 40(1): 118-137.  https://doi.org/10.1080/01431161.2018.1506184.

This paper assesses the importance of DEM source and DEM resolution for accurately mapping wetlands and draws similar conclusions to your study.

Author Response

I would like to commend the authors on a well-executed study. This was one of the better manuscripts that I have been asked to review over the past few years. Below are some suggestions.

Response: Thanks very much for your positive comment on the quality of our paper.

I would consider adding a brief discussion of how variable importance is calculated using gradient boosting. Since this method was used to assess the importance of your input variables, a discussion would be useful to improve the interpretability of the results.

Response: Thanks for your comment. We added a paragraph as

“For each of the five models, the variable influence for the decision tree ensembles was computed based on the decision trees influences [60] using the method proposed by [61]. To facilitate comparison, we scaled the total influence to be 100 and calculated the relative importance of every variable”.

Can you provide a brief justification for the terrain variables used? Given that there are a wide variety of terrain variables that can be derived from a DEM, it is important to not why these specific variables were chosen.

Response: Yes, there are currently a variety of topographic variables extracted from DEM to describe various aspects of a site such as topographic complexity, spatial heterogeneity, and landscape position. Many of these variables are highly similar and correlated. We selected those variables that are likely affect inundation and not highly correlated – the justification was added in the text.

Can you elaborate further on how the training and validation samples were collected? Based on your description, my interpretation is that wall-to-wall vegetation maps were available for your study area extent. Pixels within the mapped extents were then divided into training and validation sets using stratified random sampling. Is this correct? Also, does each sample represent a 30 by 30 meter Landsat pixel?

Response: using the “create random points” tool in ArcGIS, we first generated 10,000 random points from each vegetation class (total of 9) with the constraint of 30 m apart. As the size of some of the veg class is too small for 10,000 points, we resulted a total of 57,005 points. We then used stratified random sampling (to ensure every class of vegetation is sampled) to split the 57,005 points into training (25%) and testing (75%) dataset. 

We revised the text so that the description was clearer.

What resampling method was used to resample the LiDAR-derived DEM data?

We used bilinear interpolation (added in the text).

Would it be possible to provide the equations for the terrain derivatives used here?

Response: The details of the variables was detailed in the references.

For the cost distance surface, would it make more sense to use topographic slope as a measure of cost or impedance as opposed to elevation? Can you explain why elevation was used as the measure of impedance here?

Response: Slope might indicate the flow direction and velocity. In the floodplains of semi-arid Australia, which are very flat, the distance to source water (that the streams) is more important for a place to get inundated. The elevation-weighted distance to streams takes both distance and impedance into account.

Can you explain what monthly ranges are included in each of the four seasonal composites?

Response: Spring (Sep -Nov), Summer (Dec – Feb) Autumn (Mar – May), Winter (June -Aug)

The transformation of the time series data to polar coordinate space needs further explanation. I have not worked with these methods, and I assume that other readers may also not be familiar with these methods. Could you provide further elaboration regarding these measures and their applications in remote sensing? Please also clarify what ranges of dates were used in the analysis.

Response: we provided a detail description on how the fractional cover was converted into polar coordination system as:

“The time series (1987–2013) of seasonal fractional cover images were used to develop metrics that characterised the variation in green and non-green vegetation. First, individual fractional cover image, which encloses the proportion of green vegetation, non-green vegetation and bare soil within each pixel [55], was converted into a polar coordinate system, and the fraction of green and non-green vegetation was expressed as distance from the origin to the reference point and the polar angle (θ) between the projected line from the origin to the reference point  and the  positive horizontal axis (green vegetation) (Figure 2). Second, the time series pixel values of distance, angle and hop length (defined as the distance between two successive points in the polar space) was used to produce a range of statistics.” 

Since balanced accuracy is used in the accuracy assessment, I would explain this measure in the 2.5 Accuracy assessment and model comparison section.

 Response: Thanks. The term is briefly explained. We revised the paragraph as

“We report four indicators to compare the performance of the models: OAA, balanced accuracy (BA), Kappa coefficient of agreement, and AUC (Area Under the ROC Curve). As the dataset is highly unbalanced, BA, which is calculated as the mean of Sensitivity and Specificity, and conceptually defined as the average accuracy obtained on all classes, might provide a better indicator of the model performance”.

Did your study provide any insight as to what seasonal composite was most useful in this classification? I have read that spring data is commonly useful for wetland mapping. Do your results suggest this also?

Response: No. As the polar fractural indices mixed all seasonal data together, the algorithm developed by the vegetation team in our organisation doesn’t allow to get the makeups of the indices.

Some of your boxplot figures use two decimal places for Kappa while others use one. I would suggest using two decimal places in all figures for consistency.

Response: Thanks for our comments. Changed to two decimal places.

Would it be possible to include at least one error matric? This would be helpful for the reader to understand the sources of confusion and would be most interesting for the classification involving the largest number of classes. Would it be possible to change the axis labels for the Figure 5 importance graphs to provide more descriptive names for the variables?

Response: The description of some of the variables are quite long, it’s impractical not to use acronym.

I would recommend including the following citation:

Maxwell, A.E., T.A. Warner, and M.P. Strager, 2016. Predicting palustrine wetland probability using random forest machine learning and digital elevation data-derived terrain variables, Photogrammetric Engineering & Remote Sensing, 82(6): 437-447. https://doi.org/10.1016/S0099-1112(16)82038-8.

This paper assesses the value of different terrain variables for mapping palustrine wetlands. It also notes the importance of cost distance as distance from streams weighted by slope in mapping palustrine wetlands.

I would also recommend including the following citation:

Maxwell, A.E., and T.A. Warner, 2019. Is high spatial resolution DEM data necessary for mapping palustrine wetlands?, International Journal of Remote Sensing, 40(1): 118-137.  https://doi.org/10.1080/01431161.2018.1506184.

 This paper assesses the importance of DEM source and DEM resolution for accurately mapping wetlands and draws similar conclusions to your study.

Response: Thanks for the suggestions. The two papers are indeed highly relevant and are cited in the revision.

Reviewer 3 Report

Interesting work, thank you.

Author Response

Interesting work, thank you

We really appreciate your positive comments.